# Plant Virus-Derived Vectors for Plant Genome Engineering

**DOI:** 10.3390/v15020531

**Published:** 2023-02-14

**Authors:** Muhammad Arslan Mahmood, Rubab Zahra Naqvi, Saleem Ur Rahman, Imran Amin, Shahid Mansoor

**Affiliations:** 1Agricultural Biotechnology Division, National Institute for Biotechnology and Genetic Engineering (NIBGE), College of Pakistan Institute of Engineering and Applied Sciences (PIEAS), Jhang Road, Faisalabad 38000, Pakistan; 2Department of Biological Sciences, University of Sialkot, Sialkot 51310, Pakistan; 3International Center for Chemical and Biological Sciences, University of Karachi, Karachi 74000, Pakistan

**Keywords:** genome engineering (GE), CRISPR-Cas system, plant virus, geminivirus, viral vectors

## Abstract

Advances in genome engineering (GE) tools based on sequence-specific programmable nucleases have revolutionized precise genome editing in plants. However, only the traditional approaches are used to deliver these GE reagents, which mostly rely on *Agrobacterium*-mediated transformation or particle bombardment. These techniques have been successfully used for the past decades for the genetic engineering of plants with some limitations relating to lengthy time-taking protocols and transgenes integration-related regulatory concerns. Nevertheless, in the era of climate change, we require certain faster protocols for developing climate-smart resilient crops through GE to deal with global food security. Therefore, some alternative approaches are needed to robustly deliver the GE reagents. In this case, the plant viral vectors could be an excellent option for the delivery of GE reagents because they are efficient, effective, and precise. Additionally, these are autonomously replicating and considered as natural specialists for transient delivery. In the present review, we have discussed the potential use of these plant viral vectors for the efficient delivery of GE reagents. We have further described the different plant viral vectors, such as DNA and RNA viruses, which have been used as efficient gene targeting systems in model plants, and in other important crops including potato, tomato, wheat, and rice. The achievements gained so far in the use of viral vectors as a carrier for GE reagent delivery are depicted along with the benefits and limitations of each viral vector. Moreover, recent advances have been explored in employing viral vectors for GE and adapting this technology for future research.

## 1. Introduction

Genome engineering (GE), also known as programmable nucleases, is opening a new window to a vast array of scientific and technological possibilities. The major benefit is that it modifies the specific sequence of DNA of an organism in a targeted and precise manner, thereby increasing the efficiency of gene disruption, insertion, and correction, thus offering remarkable reproducibility [1]. The diversity of GE techniques offers the introduction of modifications into plants to obtain desirable trait(s) that define the next generation of plant breeding [2].

Decades of research in GE have culminated in the development of four major classes of sequence-specific nucleases (SSNs) that can specifically bind with the specific genomic region and induction of modification occurred in the target gene, hence referred to as “designer nucleases”. These are meganucleases or engineered homing endonucleases, zinc-finger nucleases (ZFNs), transcription activator-like effector nucleases (TALENs), and clustered regularly interspaced short palindromic repeats (CRISPR)-CRISPR associated 9 (CRISPR/Cas9) systems. These SSNs generate double-stranded DNA breaks (DSBs) at target sites and precise modifications in the genome are achieved via cellular repair mechanism [3]. The SSNs (meganucleases, ZFNs, and TALENs) recognize the specific sequences targets protein-DNA interactions, while the CRISPR-Cas system target the sequences via Watson–Crick base pairing, depending on the homology between the programmable “guide” RNA and target DNA (Figure 1A). It is a widely used system for plant genome editing because of its low cost, high efficiency, and simplicity [4]. It has been used in three site-directed nucleases (SDN) modes. The most exploited CRISPR modality is SDN-1 in which a gene is silenced and produced a phenotype with loss-of-function, while SDN-2 carried out targeted variations in the DNA deploying a template that allows a gain-of-function for the gene of interest. SDN-3 module is comparable with *Agrobacterium tumefaciens* since it permits entire gene insertion into the host DNA aided by an adequate template (Figure 1B). The SDN-2 modality requires a thorough knowledge of how specific genome mutations impact the functionality of the target [5].

The DSBs can get repair by two main avenues: (1) nonhomologous end joining (NHEJ), and (2) homology-directed repair (HDR). The NHEJ occurs when the DNA repair pathway ligates two random strands together, along with the insertion or deletions (indels) of a few nucleotides to those breaks, and can occur during any phase of the cell cycle, whereas HDR uses longer sequence homology to repair the DNA lesions, and it relies on the presence of a sister chromatid during late S phase and G2 phase of the cell cycle [6]. In NHEJ, the user-specific sequence is mostly the codon sequence of protein, which causes indels an early stop codon in the process of synthesizing a non-functional and truncated form of the target protein. Alternative approaches, such as reverse transcriptase-mediated prime editing and deaminase-mediated base editing tools of genome editing, involve neither the formation DSBs nor need donor DNA. These CRISPR gears tempt precise gene editing and turn out to be more effective than HDR repair in plants (Figure 1C,D). Following the development of adenine base editor (ABE) and cytosine base editor (CBE), dual base editor and base editing precise DNA deletion methods were primely developed in plants. As it develops base substitutions and short insertions and deletions at a relatively wide range of positions, it is not substantially constrained by its protospacer adjacent motif (PAM). Prime editing systems were developed and evaluated in rice and wheat and were shown to develop all 12 substitutions, multiple base substitutions, insertions, and deletions in rice and wheat [7,8].

One of the major challenges of CRISPR-Cas9 technology is the choice of vectors that need to be modified systematically to deliver CRISPR-Cas reagents in the plant genome. Traditional approaches for the delivery of editing components rely on the transformation protocols (*Agrobacterium* and Biolistic) or transient delivery to the protoplasts. However, both techniques are time-consuming, laborious, and also raise some legal concerns [9]; thus, advanced and reliable techniques are direly needed. One such technique could be the use of plant virus-derived vectors which have already been used for multiple purposes including the production of useful proteins [10], etc. Currently, these plant viruses are regarded as natural masters of in vivo CRISPR-Cas delivery as compared to the other delivery methods [11] due to their ability to autonomously replicate in the host genome, provides alternative means to deliver GE reagents into the plant cells. Among all the plant viruses, DNA viruses, such as wheat streak mosaic virus (WSMV) and barley stripe mosaic virus (BSMV), are used for delivery in monocots [12], while RNA viruses like tobacco rattle virus (TRV) are used for delivery in dicots [13]. Furthermore, geminivirus-based replicons could offer an almost all-rounder solution to all these limitations of GE. For example, the transcriptional analysis of *Arabidopsis* plants infected with geminivirus cabbage leaf curl virus (CaLCuV) revealed that the viral infection induces the expression of several genes linked to repairing DSBs and DNA synthesis, including DSS1 (I), DSS1 (V), DMC1, POLD3, RPA1A, and RPA1E [14]. However, there is no comprehensive study available regarding the delivery of GE delivery in plants via viral vectors. In this regard, the present review highlights the recent advances and the future of viral vectors for GE tool (CRISPR/Cas) delivery for plant improvement.

## 2. Ways to Deliver CRISPR/Cas Reagents to Plants

Owing to the new CRISPR/Cas-based gene editing technology, there has been a resurgence of interest in optimizing transformation and regeneration techniques to facilitate the rapid development of gene-edited crop lines. The delivery systems, such as *Agrobacterium*-mediated transformation and biolistic/particle bombardment, have been utilized for plant transformation, each with their own benefits and drawbacks [15,16]. *Agrobacterium*-mediated delivery and insertion of T-DNA often result in a single copy integration, while biolistic-based delivery has been shown to result in multiple integration or the insertion of partial DNA fragments, with fewer constraints on the target species [17]. Different delivery systems for GE in plants have been previously reviewed [18,19], and some of the major systems, such as non-viral (*Agrobacterium*-mediated transformation, biolistic transformation, nanomaterial based delivery), and viral based delivery systems have been discussed in this review. We also summarize studies that have used different non-viral delivery methods for genome engineering in Table 1. It is notable that editing efficiencies described in these studies vary greatly among different plant species, delivery methods, and sample types (e.g., protoplasts versus regenerated calli or plants).

### 2.1. Agrobacterium-Mediated Transformation for Genome Engineering

*Agrobacterium tumefaciens* is a Gram-negative phytopathogen that is ubiquitous in the soil [20]. It chemotactically infects the injured parts of most dicotyledonous plants under natural conditions and produces crown gall tumors [21]. The tumorigenic properties of oncogenes are mediated by the transfer (T)-DNA region of tumor-inducing (Ti) plasmids, which are approximately 160–240 kb in size, with the T-DNA region being 15–30 kb, and the virulence (vir) gene region being almost 36 kb [22]. The T-DNA comprises three sets of genes, of which two are referred to as oncogenes and synthesize mitogens (auxin and cytokinin), promoting the unrestricted growth and division of plant wound tissues to form crown gall tumors; the third set is opine (rare amino acids), producing octopine, nopaline, agropine, and succinamopine, which are utilized as carbon and nitrogenous sources by Agrobacterium [23]. The vir region encodes for proteins that initiate, process, mediate, and integrate T-DNA into the plant nuclear genome [24]. Scientists have mimicked this ability of *Agrobacterium tumefaciens* to deliver gene/s of interest into plant genome/s (Figure 2) for improving different trait(s) [25]. Agrobacterium-mediated transformation is a method of choice for genetic engineering of plants and for GE.

For GE, the genes encoding the RNA-guided endonuclease CRISPR/Cas9 are cloned onto the transfer DNA (T-DNA), integrated into the plant via *A. tumefaciens* (Figure 2), and several important trait(s) have been improved with ease [19,26]. The *Agrobacterium*-mediated transformation has been used to deliver the nucleases machinery; however, there are still a lot of limitations, such as low efficiency. To improve this, the application of ternary vectors system with some regulators, [27] such as the overexpression of *Baby boom* (*Bbm*) and maize *Wuschel2* (*Wus2*) genes, has been proposed that enhanced the transformation of *Agrobacterium* [28,29]. However, there is still a gap in the use of *Agrobacterium*-mediated transformation for the delivery of CRISPR-Cas9 systems in plants that must be overcome, or some alternative methods need to be applied. Efforts have been made for the optimization of *Agrobacterium*-mediated transformation for GE of multiple plants, such as cotton, wheat, tomato, Maize, soybean, and rice [17,30,31,32,33,34,35,36]. The *Agrobacterium*-mediated delivery of CRISPR-Cas9 has been shifted to other field crops, such as Hemp [37]. These modifications are either in CRISPR vector systems, explant selection, media composition, or the addition of some regulators [35,36,38,39]. Recently, to enhance the efficiency of *Agrobacterium*-mediated transformation for CRSIPR-Cas9, scientists have targeted the *Agrobacterium* genome [40,41] however, the technology faces multiple challenges that need to be improved further. Most recently some more advancements occurred in *Agrobacterium*-mediated transformations [26,42,43], in order to overcome these limitations, such as transgene-free editing and improvement in the delivery systems in plants [35,43,44]. Still there are some limitations of *Agrobacterium*-mediated transformation, such as the transformation of plants is not always successful as intact-transgenes insertion at the target site, but it might be escorted by re-organizations of the DNA-cassettes and genomic DNAs of the corresponding plants [45]. *Agrobacterium*-mediated transgenics also arose several biosafety concerns, such as: (1) undesired integration of T-DNA and transformation of plant [46], (2) potential gene flow from transgenic plants to the closely related plant species [47], (3) possible transfer of the selectable marker genes to the host [48], and (4) horizontal gene transfer from *Agrobacterium* to non-target plant species and avirulent *Agrobacterium* species (or other microbes) [49] which must be addressed properly. In addition, the corresponding binary-vector support sequences in the transgenic produced using *Agrobacterium*-mediated protocol arose regulatory issues, transgene/s silencing, and probable conjugative transmission [50]. The existence of switchable antibiotic/herbicide to marker-genes, a major biosafety concern globally, to sort out the transformed organisms can cause erratic changes in the transgenic organisms. Marker-free peanut transgenics were developed using cotyledon as an explant [51]; however, the most well-organized technique was introduced by Bhatnagar et al. [52]. Lately, an inclusive protocol for commercial non-marker assisted production of transgenic-rice was introduced, which is much cost-effective and labor-saving [53].

### 2.2. Biolistic Transformation for Genome Engineering

In the biolistic transformation of different plant species, the coated DNA with gold particles is delivered at high speed. After being delivered into the cell nucleus the gene of interest is integrated into their genome (Figure 2), as described earlier [54,55,56]. Furthermore, the biolistic transformation is mostly used for plants that are recalcitrant to tissue culture [57]. With this technique, multiple crops have been engineered, such as wheat, maize, and cotton [57,58]. Recently, the biolistic transformation has been used to target pollens by using CRSIPR-Cas9 systems [59]. The biolistic transformation for GE has been previously extensively reviewed [60]. However, the biolistic transformation needs a lot of effort to be established for CRISPR based delivery system in plants. Although biolistic transformation is fast and reliable, but, at the same time, this technique has multiple limitations such as low copy number and integration in multiple sites of plant genome [61,62]. Moreover, the efficiency is very low as compared to other delivery systems and very costly [63].

### 2.3. Nano-Particle Based Delivery Systems for Genome Engineering

Nanomaterials-based delivery of CRISPR reagents has revolutionized the genetic transformation of multiple crops [64]. There are multiple types of nanomaterials used for delivery systems, such as carbon nanotubes, carbon dots, graphene, graphene oxide, magnetic nanoparticles, liposome-like nanoparticles, etc. These delivery systems have been reviewed previously [65,66,67] with additional advancements, such as cationic based delivery of CRIPSR reagents for stable genome editing in plants [68]. The nanoparticle-based delivery systems have multiple advantages over other delivery methods, such as having low toxicity, choice for delivery of all biomolecules, and no dependency on plant species. Thus, it is the choice for CRISPR delivery systems in all plants species which are resistant to *Agrobacterium* or recalcitrant to tissue culture.

However, there are multiple challenges regarding the use of nanomaterials-based delivery systems for CRISPR reagents in plants, such as limited nanocarriers affected by carriers physical and chemical properties [65], cell barriers (cell wall/cell membrane, etc.), and its targeted delivery. Thus, to tackle these problems, advancements in nanomaterials are direly needed for CRISPR reagents delivery in plants.

### 2.4. PEG-Mediated Reagent Delivery for Plant Genome Engineering

Polyethylene glycol (PEG)-mediated plant transfection is a convenient tool to deliver foreign DNA or genes into protoplasts. This method has been efficaciously utilized in several plants including soybean, maize, wheat, or rice [69]. CRISPR machinery can be transferred as ribonucleoproteins (RNPs) to the plant cells by using particle bombardment and PEG-mediated protoplast transfection and lipofection. The prerequisite for the PEG-mediated transformation of protoplast for the introduction of RNPs entails cell wall removal by using pectinase and cellulase. Editing efficiency is tested using PEG-mediated transfection of rice (*Oryza sativa*), *Arabidopsis thaliana*, tobacco (*Nicotiana tabacum* and *N. attenuata*), and lettuce protoplasts. The PEG-mediated regeneration of lettuce bin2 mutant plants showed an editing efficiency of up to 46% at the targeted site without any off-targets identified in plants and 71% editing efficiencies were achieved in *Arabidopsis* protoplasts [70]. Rice zygotes were produced using PEG-mediated protocol to target GENERATIVE CELL SPECIFIC-1 (GCS1), DROOPING LEAF (DL), and GRAIN WIDTH 7 (GW7) genes which rendered 13.6–14.3%, 21.4%, and 64.3% editing efficiencies, respectively [71]. To date, protocols for regenerating a whole plant from protoplasts have been developed for numerous species, such as lettuce, potato [72], tobacco, cabbage, maize, and rice. However, PEG protocols remain limited to the plants in which protoplast isolation is established. Additionally, the regeneration methods for several plant species are quite challenging or laborious and transfection efficiencies remain poor [27]. Furthermore, the regenerated plants contain somaclonal variations and may have unstable genomes [73], raising concerns about phenotypic changes and undesired mutations. Nonetheless, for the species where the whole plant cannot be generated through this method, transient assays exploring the CRISPR system can still be effectively performed.

**Table 1 viruses-15-00531-t001:** Summary of CRISPR/Cas9 machinery used for plant genome engineering.

Transformation Method	Species	Target Genes	CRISPR System	Plant Material	Editing Efficiency	References
Protoplast transformation						
PEG	*Arabidopsis thaliana*	*Allene oxide cyclase (AOC)*	Cas9	protoplast	16%	[70]
		BRASSINOSTEROID INSENSITIVE 1 (BRI1)	Cas9 with two gRNAs simultaneously	protoplast	54–71%	
PEG	*Brassica napus* cv. Topaz	*Phytoene desaturase (PDS)*	Cas9	Protoplast	0	[74]
		*FRIGIDA (FRI)*			0	
PEG	*Brassica oletacea* var. *capitata f. alba*)	*PDS*	Cas9	Protoplast	0.14–1.33%	[74]
		*FRI*			0.09–2.25%	
PEG	*Brassica oletacea* var. *capitata f. alba*)	*GIGANTEA (GI)*	Cas9	Protoplast	2%	[75]
PEG	Hot pepper (*Capsicum annuum* cv. CM334)	*Mildew locus O 2 (MLO2)*	Cas9	Callus and protoplast	0.2% and 17.6%	[76]
	Sweet pepper (*C. annuum* cv. Dempsey)			Leaf protoplast	0.5–11.3%	
PEG	Rice (*Oryza sativa*)	*P450*	Cas9	Protoplast	19%	[70]
		*DWDI*			8.4%	
PEG	Wild tobacco (*N. attenuata*)	*Phytochrome B*, *PHYB*	Cas9	Protoplast	44%	[70]
PEG	Garden petunia (*Petunia × hybrida*)	*Nitrate reductase (NR)*	Cas9 Four gRNAs	Protoplast	5.30–17.83%	[77]
PEG	Rice (*O. sativa* cv. Nipponbare)	*DsRed2*	Cas9	Zygotes produced by gamete fusion	25%	[71]
		*DROOPING LEAF (DL)*			13.6–14.3%	
		*GRAIN WIDTH 7 (GW7)*			21.4%	
		*GENERATIVE CELL SPECIFIC-1 (GCS1)*			64.3%	
*Agrobacterium*-mediated and Particle bombardment-mediated delivery						
Particle bombardment	Rice (*O. sativa* cv. Nipponbare)	*PDS*	Cas9 with the plasmid encoding *hygromycin phosphotransferase (hpt)*	Scutellum derived embryos	3.6%	[78]
			HiFi Cas9 with the plasmid encoding *hpt*		8.8%	
			Cas9 D10A with two gRNAs and the plasmid encoding *hpt*		0	
			Cas9 with two gRNAs and the plasmid encoding *hygromycin phosphotransferase (hpt)*		62.9%	[79]
Particle bombardment	Bread wheat (*Triticum aestivum* cv. Kenong 199)	*Grain width and weight 2 (GW2)*	Cas9	Immature embryo	2.2% (TaGW2-B1) 4.4% (TaGW2-D1)	[80]
Particle bombardment	Wild tobacco (*N. tabacum* cv. Bright Yellow 2)	*ppor-RFP*	Cas9	BY2 cells	3%	[81]
Particle bombardment	Bread wheat (*T. aestivum* cv. YZ814)	*GW2*	Cas9	Immature embryo	1.3%	[80]
		*GASR7*			1.8%	
Particle bombardment	Maize (*Zea mays*)	*Male fertility gene (MS45)*	Cas9 RNP with DNA vectors encoding “helper genes” cell division-promoting transcription factors (maize *ovule developmental protein 2 [ODP2]* and maize *Wuschel [WUS]*) and selectable and visible marker genes (MOPAT-DSRED fusion)	Immature embryo	47% (28% monoallelic mutations; 19% biallelic mutations)	[58]
		*Acetolactate synthase (ALS2)*	Cas9 RNP with DNA vectors encoding helper genes; 127 nt single-stranded DNA donor		~2–2.5% (all monoallelic mutations)	
		*MS45*	Cas9 RNP only		4.0% (3.1% biallelic mutations)	
		*Male fertility gene (MS26)*	Cas9 RNP only		2.4% (0.3% biallelic mutations)	
		*Liguleless 1 (LIG)*	Cas9 RNP only		9.7% (0.9% biallelic mutations)	
*Agrobacterium*-mediated	*A. thaliana*	*AtPDS3*	CRISPR components	Leaf	37.7–38.5%	[82]
	Tobacco (*N. benthamiana*)	*NbPDS*			1.8–2.4%	

## 3. Plant Viruses and Their Role in CRISPR Reagents Delivery

To date, the most common system used to deliver the CRISPR-Cas reagents and obtain transgenic plants are the *Agrobacterium* and biolistic-mediated protocols. In addition to *Agrobacterium*-*tumefaciens*, another species *Agrobacterium rhizogens* (which can cause hairy root disease) has also been used to carry the CRISPR-Cas machinary ensuing the stable integration of foreign DNA in soybean and other important plant species with editing efficiency up to 95% [83]. However, it requires the regeneration of whole plants from roots which can be problematic for some plant species. *Agrobacterium* and biolistic-mediated delivery have several advantages, such as cheap, required technology available in most laboratories and allowing multiplex editing as multiple binary vectors can be delivered into the bacterium and co-transformed into the host cells. Additionally, it can further be used in transient assays that result in non-transgenic plants and a lower number of off-target sites edited plants. The other commonly used method is particle bombardment which consists of coating metallic microprojectiles (gold, silver, or tungsten) with DNA constructs and firing into the host plant cells at high pressure resulting in the targeted mutation in several plant species [84].

Plant viruses have been used as heterologous gene expression vectors since the beginning of genetic engineering. The advent of molecular biology and high-throughput sequencing technologies have enabled the manipulation of the viral genome to express heterologous proteins and RNAs in plants. Several recent studies have highlighted the potential use of plant virus vectors as transient delivery vehicles for CRISPR-Cas reagents in plants [85]. To date, the best alternative approach to deliver the CRISPR-Cas reagents into the plant cells is the plant viruses. Recent development in GE technologies have urged scientists to incorporate viral vectors and utilize them for the efficient delivery of GE reagents in plant cell (Table 2). The use of plant viruses as a delivery vehicle will be discussed in the following section.

### 3.1. Genome Editing by Geminiviruses: How They Can Help?

Geminiviridae, the largest virus family consists of circular, single-stranded (ss) DNA viruses infecting a wide variety of hosts ranging from staple to fiber crops worldwide, such as cotton, maize, wheat cucurbits, tomato, and several ornamental and weed plants [86,87,88], and currently pose a serious threat to the global food security. According to the International Committee on Taxonomy of Viruses (ICTV), the family, Geminiviridae, is one of the largest groups of plant viruses containing almost 485 species. Geminiviral genomes are highly reduced in size, ranging from ~2.7 to 5.5 kb encoding four to eight functional proteins (Figure 3A) present in both sense and complementary sense strand [89]. The insect pests which are involved in transmitting geminiviruses are whitefly (*Bemisia tabaci*) [87] and leafhoppers. Geminiviruses-derived vectors have been extensively used in the production of proteins, vaccines, and in inducing gene silencing in functional genomic methods [90]. In the perspective of GE, geminivirus-based replicons have attracted much attention and proved successful [91] for genome-editing technologies.

The following remarkable properties of geminiviruses style them as suitable vectors for plant genome engineering: (1) having the ability to make infection in a wide range of plants belonging to various species at once; (2) requiring a very smaller number of proteins for initiation of replication inside hosts (in case of mastreviruses; replication associated protein: Rep); (3) its expression is regulated by its own natural promoter present in the intergenic region and any user-specific inducible/constitutive promoters [91]; (4) independently replicate inside the host by homologous recombination (HR)-dependent replication, it reverts the host cell into S phase fit for HR if associated with SSN and complementary target sequences [92]; and (5) it produce a large amount of amplicons via replication inside the host cell, in turn developing a high number of SSNs and target sequences when used as a vector for genome engineering thereby increasing targeting efficiency.

The engineering of geminiviruses as vector has been used for the heterologous protein expression in plants [90]. However, the cargo capacity of these geminiviruses is restricted because they can convert themselves into non-infectious replicons by replacing their functional genes that are important in the initiation of infection and cell-to-cell movement with heterologous sequences, for example, SSN expression cassettes and repair templates. To overcome this, the coding sequences of movement protein (MP) and coat protein (CP) of geminiviruses have been removed, thereby eliminating the possibility of plant-to-plant insect transmission and cell-to-cell movement. The absence of CP caused a high number of dsDNA replicons intermediates production. Without CP, the packaging and sequestering of ssDNA into virions is not carried out, thus the interaction of CP and Rep is lost which ultimately represses the viral replication. In the seminal work by Baltes et al. (2014), it was demonstrated that geminiviral replicons can be successfully utilized to deliver SSNs (ZFNs, TALENs, or CRISPR-Cas), and to produce a high amount of repair template in plant cells (Figure 3B). The work carried out by Baltes demonstrates that geminiviral replicons promote gene targeting in two ways: (1) they allow the replication of the repair template in high copy numbers, and (2) Rep/RepA favors this mechanism through unspecified pleiotropic effects [91].

Baltes et al. (2014) described the geminivirus (Bean yellow dwarf virus; BYDV) based replicons as a vector for transient expression of ZFNs, TALENs, and CRISPR/Cas system along with the delivery of DNA repair templates. The virus has a considerable cargo capacity, therefore could deliver these SSNs. The authors developed a deconstructed version of BYDV for the efficient delivery of these SSNs and a repair template in tobacco cells and achieved gene targeting at a specific integrated reported gene resulting in the efficient HDR thereafter. They further explained that DNA carried geminiviruses can be used as a template for homologous recombination and this technology has been successfully used to produce calli and plantlets with precise DNA sequence modifications [16]. Gil-Humanes et al. (2016) developed a replicons-based system by using a deconstructed version of the wheat dwarf virus (WDV) to engineer cereal crops. The replicons have successfully achieved a 110-fold increase in expression in a reporter gene compared to the non-replicating control. Furthermore, replicons carried repair templates and SSNs (CRISPR/Cas9) have achieved gene editing at a ubiquitin locus in all the three homeoalleles (A, B, and D) of wheat at 12-fold greater frequencies compared to the non-viral delivery method. The findings further confirmed that WDV-based replicons make it possible to edit complex cereal genomes without integrating the gene editing reagents into the plant genome [93]. Cermak et al. (2015) employed geminivirus-based replicons to generate heritable modifications at a 10-fold higher frequency compared to the traditional delivery methods (i.e., *Agrobacterium*) to the tomato genome by inserting promoter to the upstream of a gene that regulates the anthocyanin biosynthesis. The resulting gene editing is the overexpression and ectopic accumulation of the anthocyanin biosynthesis pigments in tomato tissues and the targeted modifications was transmitted to the next progeny in a Mendelian manner. In addition, they conclude that by employing geminivirus based replicons, high-frequency, precise modification in the tomato genome was achieved, and it can also overcome the efficiency barrier that made gene targeting in plants more challenging [94]. Wang et al. (2017) carried out gene editing by employing CRISPR/Cas9 through WDV-based replicon system and provides a simple and efficient tool for the delivery of abundant donor DNA into rice cells, resulting in an HDR efficiency of up to 19.4% in transgenic rice plants. Lastly, two studies used similar approach for the targeted gene editing in potato crop [95,96] and overcame three essential barriers: (1) employing geminivirus-based replicons for GE in plants, (2) HDR efficiency increasing in plants, and (3) geminivirus-based HDR used for the development of permanent transgenic lines.

Oh et al. (2021) and Zhang et al. (2022) reported an approach in which they used plant viruses to deliver Cas proteins and guide RNAs (gRNAs) into a plant cell without a complicated experimental procedure. This strategy is known as virus-induced genome editing (VIGE) (Figure 3C), and has been lauded as a game-changer in CRISPR-based genome editing due to several benefits over conventional delivery methods [97,98,99]. This system can be used for generating knock-out libraries as an alternative to the conventionally used virus induced gene silencing (VIGS), which causes non-specific silencing especially for highly homologous genes. VIGE has emerged as a powerful tool with multiple benefits including high editing efficiency, accuracy, operability, and simplified protocol for the development of gene-edited DNA-free plants. In addition, VIGE is also useful for avoiding various problems caused by *Agrobacterium*-mediated integration of T-DNA harboring CRISPR reagents, VIGE will reduce insertional mutation rates, off-target effects, and cross-contamination of DNA between wild-type and transformed plants. 

The deconstructed version of cabbage leaf curl virus vector has been used for the expression of gRNAs in Cas9 expressing stable transgenic lines and successfully induced the systemic genetic mutation in plants. This was the first report in which geminivirus-based vector have been used and induce targeted editing of the endogenous genes of the *N. benthamiana* [100]; thus, the VIGE-based on CRISPR-Cas system can be used to study individual gene function. Together, VIGE tool is an effective approach that enables genome engineering in many plants. 

### 3.2. RNA Viruses a Potential Delivery System In-Planta

The use of DNA viruses creates a possibility for inserting foreign DNA into the plant genome while RNA viruses provide an advantage over DNA viruses because the infectious cycles take place in the host cytoplasm, thus resulting the plants free from foreign DNA, which also avoids raising ethical issues and regulatory concerns. Numerous plant RNA viruses including tobacco rattle virus (TRV), tobacco mosaic virus (TMV), pea early browning virus (PEBV), barley strip mosaic virus (BSMV), foxtail mosaic virus (FoMV) and beet necrotic yellow vein virus (BNYVV) have been used exclusively as vector for the delivery of sgRNAs into the plant cells [101,102,103,104] and editing efficiencies up to 80% have achieved. Tobacco rattle virus (genus *Tobravirus*, family *Virgaviridae*) is a bipartite, positive single stranded RNA (+ssRNA) virus that infects more than 400 different plant species belonging to 50 families. It is transmitted by nematodes (family; *Trichodoridae*), mechanically and through seed transmission. It has two genome components, TRV1 (or RNA1) which is essential for viral movement and replication proteins whereas TRV2 (or RNA2) which encodes the coat protein (CP) and many other nonstructural proteins. TRV also contains important genes encoding 134 and 194 kDa movement protein (MP), and 16 kDa cysteine-rich protein, whose function has not been identified yet. TRV2 encodes the CP and non-structural proteins involved in nematode transmission, though they are non-functional for infection cycle. Therefore, by using TRV2 as a vector, two non-structural protein encoding genes can be changed with multiple cloning sites for integrating segments of interest, and for heterologous protein expression and host genes for VIGS [105]. In addition, TRV has many advantages including: (1) vast host range (more than 400 host species) and migrating ability to the growing tissues, (2) their smaller genome size provides cloning, library construction, and multiplexing and agroinfections, and (3) their RNA genome does not integrate in the plant genome [97].

TRV-based vectors were the pioneers to deliver ZFNs and TALENs into petunia and tobacco plants which leads to the permanent and heritable genome modifications in the infected plants [106]. TRV-based vector can be used to target native genes for the generation of crop plants with novel traits. For the delivery of CRISPR-Cas reagents, Ali et al. (2015) demonstrated a TRV-based vector to successfully edit the *N. benthamiana* and *A. thaliana* genomes and the detection of targeted changes in the progeny of transgenic plants provides evidence of the TRV efficiency to infect the germline cells [13]. Furthermore, another RNA virus has also been used to deliver different nucleic acids into various plant species, such as pea early browning virus (PEBV). The PEBV-RNA2 has been modified and delivered sgRNA into *Arabidopsis* more efficiently than TRV. Another advantage of employing PEBV as vector is that it also infects meristematic tissue, which allowed for the recovery of seeds with desirable changes [107,108]. Another positive-strand RNA virus that expresses a large amount of CP from a viral promoter that can be easily modified through partial substitution of CP with heterologous genes and in multiple hosts, it allows for high-level gene expression and make it suitable for prolonged integrity. Although the deletion of CP impairs its systematic movement and high concentration of sgRNA can be delivered, that leads to the efficient editing in *N. benthamiana* lines. Continuously expressing Cas9 is the key feature for exploring the feasibility of viral vector in GE. For instance, beet necrotic yellow vein virus (BNYVV) (genus Bunyavirus) based replicons have been shown to deliver sgRNAs for efficient GE in *N. benthamiana* [104], and this approach has also been expanded for targeted mutagenesis in plants other than model plant by employing barley stripe mosaic virus (BSMV) (genus Hordeivirus) in *Triticum aestivum* and maize [109], or using *foxtail mosaic virus* (FoMV) (genus Potexvirus) in the model plant (*N. benthamiana*), maize, and green foxtail [103]. A recent report on employing BSMV harboring different sgRNA was effective in inducing concurrent editing in multiple target genes in wheat, and most of them are being transmitted to the next progeny [110]. In addition, two negative-sense single stranded RNA viruses (barley yellow striate mosaic virus and sonchus yellow net rhabdovirus) have been concurrently used to transfer Cas9 and sgRNA, and achieved systemic gene editing in the model plant *N. benthamiana* [111]. To bypass the other major limitation that virus-induced mutations cannot be passed to the next progeny, researchers cleverly used RNA mobile elements and fused them with sgRNAs and introduced them into TRV-RNA2. The *Agrobacterium*-mediated infiltration into the somatic tissues while the mobile elements have directed them into the shoot apical meristem cells, thereby introduction of heritable mutations in the next generation with 100% efficiencies [112] (Figure 3D).

**Figure 3 viruses-15-00531-f003:**
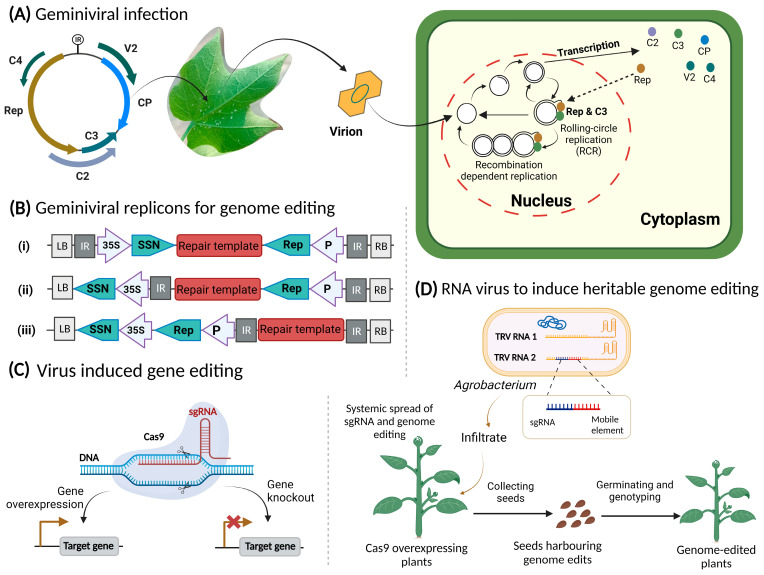
Overview of geminiviral infection and strategies for delivering CRISPR-Cas reagents. (**A**) Geminiviruses (TYLC, monopartite, in this figure) have single-stranded (ss) circular genomes (left) encapsulated into a geminate viral particle. This virion is acquired and retained by a tiny insect vector, a whitefly, which transmits it to a healthy plant in a persistent, circulative manner. Inside the plant cell, viral replication relies exclusively on host cell machinery. Recently, Wu et al. (2021) reported the role of plant DNA polymerases α and δ in the replication of the geminiviral genome. The viral ssDNA genome is converted into the dsDNA replicative intermediate by DNA polymerase α, which is then replicated by DNA polymerase δ to produce new viral ssDNA through a rolling-circle mechanism [113]. Geminiviral replication can also use a recombination-dependent replication mechanism, which is based on homologous recombination between viral dsDNA and partially replicated ssDNA molecules. The new ssDNA molecules can interact with the viral capsid protein (CP) synthesized in the cell to form new virions, which will be available for vector acquisition. (**B**) Structure of geminiviral replicons for gene editing. The T-DNA harboring left, and right borders indicate the LB and RB, respectively. SSN, sequence-specific nuclease; P, promoter (constitutive and inducible); IR, intergenic region. In (i), the Rep protein, the SSN, and the repair template sequence are included as part of the replicon; in (ii), SSN is outside the replicon and can only be expressed from the original T-DNA. In (iii), both the SSN and Rep protein both remain outside the replicon. The Rep protein does not need to be included in the same T-DNA and can instead be provided independently. (**C**) Virus-induced gene editing (VIGE) carry a single guide RNA (sgRNA) co-delivered with Cas9 or introduced into transgenic Cas9 plants. This technology has the capacity of precise gene editing, gene insertion, overexpression, or all three simultaneously. (**D**) Employing plant RNA virus to induce heritable genome editing. The complex of sgRNA fused with an RNA mobile element is introduced in tobacco rattle virus (TRV) RNA2 plasmid. Following *Agrobacterium* infiltration into Cas9-overexpressing plants, the guide RNA can be spread systemically in the plant via mobile element resulting in the inheritable mutagenesis. Created with BioRender.com.

**Table 2 viruses-15-00531-t002:** Plant viral vectors utilized for genome engineering in different plants.

Virus Type	gRNA	Nucleases Type	Plant Species	Mutation Heritability	Reference
DNA viruses					
CaLCuV	AtU6-gRNA	_	SpCas9-overproducing tobacco (*N. benthamiana*)	No	[111]
WDV	TaU6-gRNA	SpCas9	Wheat (*T. aestivum*)	No	[93]
	OsU6-gRNA	SpCas9	SpCas9-overproducing rice (*O. sativa*)	No	[114]
BeYDV	AtU6-gRNA	ZFN, TALEN, SpCas9	Tobacco (*N. tabacum*)	No	[91]
	AtU6-gRNA	TALEN, SpCas9	Tomato cv. MicroTom	No	[94]
	AtU6-gRNA	TALEN, SpCas9	Potato	No	[96]
	AtU6-gRNA	SpCas9	Potato	No	[95]
	AtU6-gRNA	SpCas9	Tomato cv. MicroTom	No	[115]
Positive (+) strand RNA virus					
TRV	_	ZFN	Petunia (*Petunia hybrida*), Tobacco (*N. Tabacum*)	No	[106]
	_	Meganucleases	Tobacco (*N. alata*)	Yes, Low frequency	[107]
	PEBV-gRNA	_	SpCas9-overproducing tobacco (*N. benthamiana*)	Yes, Low frequency	[101]
	PEBV-gRNA	_	SpCas9-overproducing *Arabidopsis thaliana,* Tobacco (*N. benthamiana*)	No	[108]
	PEBV-gRNA-FT	_	SpCas9-overproducing tobacco (*N. benthamiana*)	Yes, High frequency	[112]
PVX	BMV-gRNA-tRNA	_	SpCas9-overproducing tobacco (*N. benthamiana*)	No	[116]
	PVX-gRNA	SpCas9/AID	*N. benthamiana*	No	[117]
PEBV	PEBV-gRNA	_	SpCas9-overproducing *A. thaliana,* Tobacco (*N. benthamiana*)	No	[108]
FoMV	FoMV-gRNA	_	SpCas9-overproducing *A. thaliana,* Maize (*Zea mays*), *Setaria viridis*	No	[103]
FoMV	AtU6-gRNA	SpCas9	*N. benthamiana*	No	[118]
TMV	TMV-gRNA-ribozyme	_	*N. benthamiana*	No	[102]
BSMV	BSMV-gRNA	-	SpCas9-overproducing tobacco (*N. benthamiana*)*,* Maize (*Zea mays*), wheat (*T. aestivum*)	No	[109]
BNYVV	p31-gRNA	_	SpCas9 overproducing tobacco (*N. benthamiana*)	No	[104]
ToMV	AtU6-gRNA	Split-SaCas9	*N. benthamiana*	No	[119]
Negative (−) strand RNA virus					
BYSMV	BYSMV-gRNA	SpCas9	Tobacco *N. benthamiana*	No	[120]
SYNV	SYNV-gRNA-tRNA	SpCas9	Tobacco *N. benthamiana*	No	[111]

## 4. Pros and Cons of Viral Delivery Systems

The plant viruses can be engineered as viral vectors for the transient expression of RNAs and heterologous proteins and have served as spectacular tools for several fundamental and translational studies [97]. The plant virus-derived vectors provide numerous advantages, such as (a) easy to manipulate; (b) higher transient expression owing to high gene copy number leading to the swift development of the desired product; (c) genome can be used as repair template; (d) accumulate at high levels (repair template and sgRNAs); (e) it can spread systemically in plants resulting high expression and gene editing efficiency include expression of multiple sgRNAs from a single viral genome that allows multiple targeted gene editing (VIGE); (f) capability to screen various construct variants among diverse host plant genotypes, therefore evading poor construct and interruptions in stable plant regeneration/transformation of host plants, even in the plants those are difficult to be transformed; (g) capability to achieve spatio-temporal gene expression at different growth stages of plants by making changes in the timings of inoculation; and (h) efficient gene expression in all vulnerable plant hosts, without certain position effects among various transgenic lines [121].

Limitations of plant viral based vectors include (a) the small genome size of geminiviruses causes hindrance and impotent to transfer long DNA chunks e.g., Cas nucleases (~4.2 kb) [122]. The cargo capacity has been rewarded by deconstructed geminivirus replicons that only carry the intergenic regions (IRs) and the replication-associated proteins (Rep/RepA) essential for replication [123]. The coat protein of some bipartite begomoviruses may be replaced by the desired sequence for up to 800 bp to 1000 bp [124], though this size is not preferable for achieving the expression of site-specific nucleases, such as ZFNs, TALENs or Cas9, it is sufficient to express and produce a high amount of sgRNA; (b) only transient expression usually without any transfer of desirable characters to subsequent generations through breeding or through seed; (c) due to mutation or deletion, the introduced genes may be lost over time (more problematic with larger inserts); (d) adverse effects could be possible on the host or interactions with other viruses; and (e) transmission to the other susceptible crops or wild hosts may also be possible.

## 5. Future Outlook

CRISPR-based genome engineering is a revolution for the technological advances in the field of biotechnology. With the advent of progress in technology, now several ways for delivering the GE components have been introduced that could potentially swap or add on to the conventional delivery methods like *Agrobacterium* or particle bombardment approaches. The latest approach is the usage of plant-based viral vectors to transfer the GE machinery shows promising results to be adopted for robust GE in plants. The identification and exploitation of different viral groups have been executed so far; however, the geminiviruses harboring DNA genomes have remained under more limelight for genome engineering in plants. Geminiviruses took the credit due to having a large number of plant host ranges and the ability to efficiently cargo the GE components to the plant cells. Similarly, research has shown that RNA viruses also aid in producing GE plants with superb efficiency and robustness.

Plant viruses are naturally harmful to plants as they develop massive deadly diseases to them; therefore, the infection imparted by these viruses-based vectors might be deleterious for target plant species. Viral vectors are useful for GE in plants owing to their aggressive infection mode; however, they necessitate appropriate features and modifications for accommodating extra DNA or RNA to deliver to plants. Furthermore, though there are different methods available to predict the protein stability and folding, more methods are required to predict the interaction of individual constructs with the viral vector and to elucidate the effect of folding of the modified protein on the virus infectivity and efficacy of GE machinery transfer. The innovative next-generation sequencing and bioinformatics tools will be of great help to explore these interactions. It will still be prudent to study a wide range of viruses in the future for engineering in a way that these could be utilized for expression and of foreign proteins in different monocots and dicots. Additionally, there is a dire need to explore viral vectors for efficient and stable expression of larger or multiple proteins in the most suitable plant host system. Further investigations will extend the broad host plant range and efficient transmission to next-generation progeny with a high copy number of certain genes.

Viral vectors are usually derived from plant viruses that are transmitted by different insect pests. These insects are often host-specific, therefore restricting the virus delivery to certain hosts. Though there are some methods optimized to deliver viral vectors to plants, for example, infiltration, more efficient methods to achieve transmission to different host plant ranges can be optimized to avoid the necessity for an insect for initial infection establishment. Currently, there is some popular host plant range available that has been utilized for viral vectors but might not be safe for the purified production of certain protein due to the co-occurrence of certain unnecessary secondary metabolites. Therefore, developing viral vector assays for plants lacking undesirable metabolites will be an opportunity to yield purified products with more acceptance for commercial applications. One of the limitations to using viral vectors could be the Intellectual property or regulatory rights that might hinder their commercial application. This can be overcome by collaborations, licensing, and the spread of public awareness to facilitate the utilization of viral vectors for practical and commercial purposes.

Plant-based viral vectors are extremely versatile tools for forward and reverse genetics as they provide an opportunity to envisage plant gene expression, function, and interactional studies. In a nutshell, in recent years, the progress on viral vectors derived from plant viruses can potentially be applied to promisingly improve agriculture for the betterment of crop trait(s), particularly amid to climate change scenario. The future progress in the field of biotechnology will drive genome engineering through viral vectors in more refined modes that will open new windows for the scientific community in plant genome engineering.

## Figures and Tables

**Figure 1 viruses-15-00531-f001:**
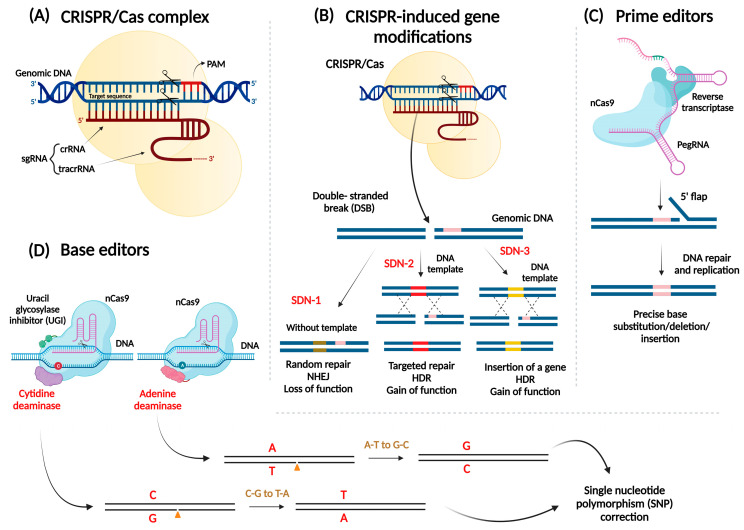
Overview of CRISPR/Cas system, the three CRISPR-induced gene modifications, and Reverse transcriptase-mediated and deaminase-mediated genome editing tools in plants. (**A**) CRISPR/Cas system, and (**B**) three CRISPR-induced modification methods, i.e., SDN-1, SDN-2, and SDN-3, (**C**) Prime editing (PE) technology that is composed of a fusion of nCas9 along with reverse transcriptase and a prime editing guide RNA (pegRNA), (**D**) Base editing technology. Cytidine and adenine deaminase is fused with nCas9 to generate a cytosine base editor (CBE) and adenine base editor (ABE), respectively. SDN: Site-directed nuclease; crRNA: CRISPR RNA; tracrRNA: trans-activating crRNA; nCas9: Cas9 nickase; HDR: homology-directed repair; NHEJ: nonhomologous end joining; sgRNA: single guide RNA.

**Figure 2 viruses-15-00531-f002:**
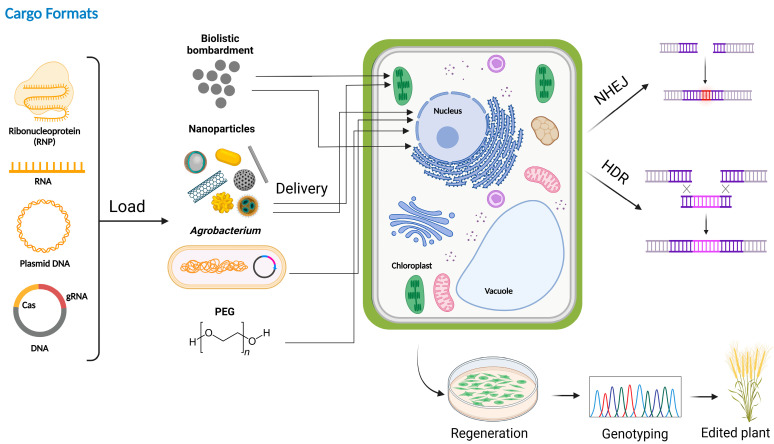
Schematic representation of CRISPR-Cas delivery methods. The ribonucleoproteins: RNPs (consists of Cas9 and an in vitro transcribed sgRNA), RNA-encoding CRISPR-Cas reagents, DNA can be delivered into the plant cells employing biolistic bombardment, a gene gun, *Agrobacterium* cells, or polyethylene glycol (PEG). The CRISPR-Cas reagents, in the plant cell nucleus, generate site-specific DSBs which might be repaired by NHEJ or HDR pathways. The uncontrolled, but predictable indels are generated in the NHEJ pathway while in the existence of a donor DNA template, DSBs most likely repaired via HDR resulting in the generation of precise modifications. Furthermore, genotyping is utilized for the identification of gene-edited plants.

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
