# Peer review of "Plant Virus-Derived Vectors for Plant Genome Engineering"

_viruses, 2023, doi:10.3390/v15020531_

Round 1
Reviewer 1 Report
Mahmood et al. wrote a review on the use of plant virus-derived vectors for genome editing, which is an important area for gene function studies and germplasm improvement. The authors first described the traditional methods for delivery of genome editing reagents and their advantages and disadvantages and then highlighted the recent advances in the use of plant virus-derived vectors for plant genome editing. The review should be of interest to researchers in plant biologists who are seeking tools for efficient modification of plant genomes. My comments are:
Major: The focus should be the use of plant virus-derived vectors for genome editing, but the authors put too much ink on other delivery methods and only section 3 on the main subject. I would like to see more details on virus-derived vector-based genome editing. How the constructs were made? What is the efficiency? Rate of off-target effect? etc. I would like to see a table of comparison of genome editing with different delivery methods and with specific numbers and references.
Minor: the manuscript needs some language work because there are many grammar errors and sentences are either lengthy or difficult to understand. For examples:
(1) line 23-24, punctuation problem, hard to understand
(2) line 34, “millions”? “many” would be better
(3) line 38, “trait/s” is usually written as “trait(s)” (same problem on a few other places)
(4) line 89-91, grammar issue/awkward expression
(5) line 102-104, in the way that is written, TRV can be confused as one of the DNA viruses
(6) line 118, “drawbacks. [14] [15]..” should be “drawbacks [14][15].”
(7) Line 122, “been reviewed, previously” should be “been previously reviewed”
(8) Line 129-132, the sentence started with “Oncogenes …” needs to be written more clearly
(9) Line 132-136, the sentence started with “T-DNA comprises …” needs to be written more clearly
(10) Line 144-148, this long sentence needs to be written as two or three shorter sentences that have clearer structure and are easier to understand.
(11) Line 158, “but still, the technology faces multiple challenges needs to be improved further.”, grammar error.
(12) Line 161, “Although …”, incomplete sentence
(13) Line 186, “Although, biolistic transformation…”, comma is not needed
(14) Figure 2 legend. “RNPs” needs to be spelled out
(15) Line 258,”4” should be “4)”
(16) Line 271-274, the sentence started with “Due to the lack of CP…” needs to be rewritten
(17) Line 275, “both, to deliver …, and to produce…”, I think the two commas just make the sentence broken and awkward, and they can be simply removed
(18) Line 351, “two genomes”, I would use “two genomic segments”
(19) Line 365-366, “successful expression of meganucleases are carried out by …”, grammar error
Author Response
Please find attached replies to comments from Reviewer 1.
Reviewer 2 Report
Comments manuscript Mahmood et al.
Mahmood and co-workers present a compilation of the most frequently used tools for genome editing in plants used to date. Along this review, they discuss about the pros and cons of each method, giving more weight to the use of plant virus-derived vector for plant genome engineering over other most traditional approaches (Agrobacterium- and bombardment-based methods) as a faster strategy to implement in the usual protocols.
Although this review would be of interest to this field, this reviewer finds that the manuscript can be substantially improved in various aspects. Among them, my major concerns are: (1) the English needs careful revision and several sections of the text need to be rewritten for better comprehension, (2) certain sections of the text are somewhat disorganized and should be restructured in order to improve reading flow. On the other hand, (3) there are inaccuracies in this review that the authors must correct as they do not properly reflect the actual information gathered in the cited references; or the way in which the authors have interpreted previous publications and consequently written this review is confusing.
Regarding English, I have indicated only some points to correct, although not all. Please check the entire manuscript again.
Some other concerns to solve:
- Line 166-167: Please indicate at least some or the most important biosafety concerns regarding Agrobacterium-mediated transformation.
- In the Figure 2, the authors summarized CRISPR-Cas delivery methods where PEG is included. However, nothing is mentioned in the text regarding the PEG-mediated CRISPR-Cas transformation. Please, include a short paragraph regarding this point.
- Line 248-249: If insect vectors are whitefly AND leafhoppers (only), then “etc” must be removed. However, if among insect vectors we find whiteflies and leafhoppers, then “and” must be removed from this sentence.
- Line 254: to the various families ïƒ to various families
- Line 330: remove parenthesis in (CaLCuV).
- Line 334-337: based on which information (previous publications, data, or similarities with previously cited examples in this manuscript – such as CalCuV or WDV) the authors propose these viruses (TGMV, ACMV, MSV, and BeYDV) and not others as a promising method? Please, state this clearly in the manuscript for a better understanding.
- Line 338: System ïƒ system
- Line 345-347: Please, rephrase. It seems that this sentence contains duplications.
- Line 349: to the 50 families ïƒ to 50 families
- Line 349 to line 363 require an extensive rewriting.
- Line 361-362: Compared to Geminiviruses, TRV genome is not smaller at all. Please, correct this sentence.
- Line 359-361: Indicate with (1) and (2) the two first advantages if corresponding.
- Line 364: into the petunia ïƒ into petunia
- Line 365: Nicotiana alata should be written in italics.
- Line 367-368: This sentence is difficult to understand. I think something is missing or at least not clear to this reviewer.
- Line 375-379: Which positive-strand RNA virus the authors refer to? PEBV? Or another one? Please, write this section more clearly.
- Line 380, 383, 387 (and others, if corresponding): N. benthamiana in italics.
- Line 382: Beet ïƒ beet
- Line 385: Barley ïƒ barley
- Line 395: TRA ïƒ TRV
- In figure 3A, the genome scheme represents monopartite geminiviruses only. Please indicate this in the figure and figure legend. Also, the genome organization selected for this panel represents monopartite begomoviruses including (as an example) the tomato yellow leaf curl virus, for which has been shown that the virus-encoded C3 may act selectively, recruiting DNA polymerase d over e to favour productive replication (Wu et al., 2021, Nature comm.). Therefore, not only Rep but also C3 participate in the DNA viral replication event. This is not properly represented in the figure. Perhaps the authors just wanted to represent the case of the monopartite geminiviruses belonging to the Mastrevirus genus. If that is the case, the authors should replace the genome organization scheme for the corresponding to mastreviruses.
If appropriate, I will be happy to review the revised version of this manuscript.
Author Response
Please find attached response to comments by reviewer 2

Reviewer 3 Report
The review article by Mahmoud et al discusses on the potential utility of virus vectors as tools for genome editing. The data presented in this review is not unique. As this is one of he emerging fields in plant research the review draws more attention to the review. Even though there are several reviews on genome editing technologies, relatively little attention was drawn towards the utility of viruses as GE tools. This review shed some light in this direction. Despite these merits, the draft in its current format is not suitable for publication. The draft needs improvements in terms of the language used and also needs better flow of information.
L24: TRV in brackets
L25: change "gears" to "systems"
L34-35: rephrase "Besides its millions of benefits, the major one is..."
L44: add "s" to nuclease (ZFNs)
L44: expand TALENS completely .... i.e efffector nucleases
L56: change N-3 to SDN-3
L95: "Regarding" replace with a better connecting word/phrase
L99: "efficient machinery" ....what machinery? be specific
L118-121: correct grammar/ tense
Section 2. This section can be much more concise. probably in one paragraph, so that readers can directly get into the virus based delivery systems
In section 3. Authors are listing the experiments carried out by different virus vector systems. But it requires comprehensive analysis in highlighting the general lackings or advantages over other systems, areas for improvements etc. with specific examples.
Author Response
Please find attached the replies to reviewer 3.

Round 2
Reviewer 2 Report
The authors have addressed all my comments/observations therefore, I do not have further concerns.
Reviewer 3 Report
In the current draft authors made considerable improvements and incorporated the changes I have made. I accept the current version for publication with the following few minor edits/ suggestions.
L22. "the different plant viral vectors, such as DNA and RNA viruses which" ...change to " the different DNA and RNA virus based plant viral vectors which"
L58. "get repair" change to "get repaired"
L97. "Among all the plant viruses"....delete "all"
L237. delete "a gene gun" or keep it in brackets
L244. Authors should think of ways in improving the presentation of data in Table 1. The current format is okay but this can be improved by keeping only informative columns.
Protoplast transformation can be changes to "PEG based protoplast transformation" and the first column and the plant material columns can be removed.
L262. how HTs technologies enabled manipulation of viral genomes?
L266. "So till to date, the best...." this is a statement with no clear basis or evidence and also alternative for what method. I suggest to remove or rephrase the statement
L433-440: Is this description needed in the figure legend.
L499: "viral vectors are beautiful for GE in plants" rephrase